# Finite Element Analysis of Cushioned Diabetic Footwear Using Ethylene Vinyl Acetate Polymer

**DOI:** 10.3390/polym13142261

**Published:** 2021-07-09

**Authors:** Mariyam J. Ghazali, Xu Ren, Armin Rajabi, Wan Fathul Hakim W. Zamri, Nadia Mohd Mustafah, Jing Ni

**Affiliations:** 1Department of Mechanical and Manufacturing Engineering, Faculty of Engineering and Built Environment, Universiti Kebangsaan Malaysia, Bangi 43600, Selangor, Malaysia; wfathul.hakim@ukm.edu.my; 2School of Mechanical Engineering, Hangzhou Dianzi University, Hangzhou 310018, China; renxu@hdu.edu.cn (X.R.); nj2000@hdu.edu.cn (J.N.); 3Faculty of Medicine, Universiti Teknologi MARA, Sungai Buloh Campus, Sungai Buloh 47000, Selangor, Malaysia; nadiamustafah@uitm.edu.my

**Keywords:** ethylene vinyl acetate, footwear finite element analysis StepEase^TM^

## Abstract

With the development of societies, diabetic foot ulcers have become one of the most common diseases requiring lower extremity amputation. The early treatment and prevention of diabetic foot ulcers can considerably reduce the possibility of amputation. Using footwear to redistribute and relieve plantar pressure is one of the important measures for the treatment and prevention of diabetic foot ulcers. Thus, the evaluation and prediction of the distribution of plantar pressure play an important role in designing footwears. Herein, the finite element method was used to study plantar pressure under two kinds of foot models, namely, the skeletal structure foot model and the whole foot model, to explore the influence of human bones on the pressure of the soles of the feet and obtain accurate foot pressure. Simulation results showed that under the two models, the plantar pressure and the pressure from the footwear with ethylene vinyl acetate were all reduced. The total deformation demonstrated a slight increase. These stresses are very useful as they enable the design of suitable orthotic footwear that reduces the amount of stress in individuals with diabetic foot ulcers.

## 1. Introduction

Retrospective and prospective research indicated that elevated plantar pressure is the major cause of plantar ulceration in the feet of patients with diabetes [1,2]. Serious plantar ulcers usually lead to amputation, which can cause severe physical and psychological traumas to the patient. Reducing plantar pressure to prevent the occurrence of diabetic foot ulcer by using footwear is currently a hot topic in research [3,4]. However, the applicable environment of footwear is complicated and diverse due to the different physical conditions of patients and the treatment situation. The mechanism of action of footwear in relieving foot pressure and preventing ulcer is not clear. However, it does play a certain protective role, which is a consensus [5,6,7]. Therefore, redistributing the plantar pressure and reducing the peak pressure of the foot are necessary. Footwear is a good choice due to its convenience [8,9]. Biomechanical analysis of the foot is the first step in the research and design work of footwear [1,10]. Several works were conducted by scholars in this aspect. However, some scholars used the finite element method for simulation analysis due to the difficulty of actual measurement, the inaccuracy of the actual acquisition, and the unrealizable measurement of internal stress [11,12,13].

A previous study [14] used 3D finite element analysis to study the effect of full-contact insole on the redistribution of plantar stress. The results showed that except for individual areas, the pressure peak and average normal stress of most plantar areas are reduced when wearing full-contact insole. The peak normal stress is reduced from 56.8% to 19.8% compared with that of flat insole state.

Taha et al. [15] performed a finite element analysis on the human foot model to study the dynamic behavior and internal load conditions of a flat ground neutral standing. They simulated the foot structure to help design the instrumented insole. They also developed a 3D finite element foot model by using the computed tomography data of bone and soft tissue structures of the human foot. They simulated the stress of the foot under different stress conditions. Their results showed the peak pressure at the first metatarsal, fifth metatarsal, and heel. Their work could assist researchers in studying plantar pressure and developing custom insoles. Cheung et al. [16] studied the internal stress/strain of the human foot and the pressure distribution at the foot support interface under loading. The existing measurement technology that uses stress sensor could measure the plantar pressure well. However, the stress/strain could not be induced in human foot. Therefore, by using the actual geometry of the foot bone and soft tissue obtained from the 3D reconstruction of the magnetic resonance image, a 3D finite element model of the human foot and ankle was developed, and relevant stress and strain studies were carried out to accurately obtain the stress/strain inside the human foot.

A previous study [16] developed a 3D finite element model of the human foot and foot complex and a custom-molded insole through magnetic resonance imaging and 3D reconstruction of surface digitization. Considering the effects of soft tissue, such as human skeletal muscle, on stress distribution, the distal humerus, tibia, 26foot bones, 72 major ligaments, and plantar fascia were embedded in a volume of soft tissue. The finite element analysis results showed that the custom-molded shape is more important than the hardness of the insole material in reducing the peak pressure of the sole. Hardness refers to the overall physical properties of the insole when the sole is compressed.

The above studies were dominated by the idea of finite element simulation, whilst others used theoretical models, which involve less actual manufacturing of actual footwear. These studies considerably facilitated the development of the present work. However, some aspects of these studies remain unconsidered. In the present work, the current industrial material of shoes (i.e., ethylene vinyl acetate, EVA) was used to evaluate StepEase™ via the finite element method to improve the manufacturing of footwear. Compared with traditional research work, this article first put forward the idea of separating the bones and soft tissues of the foot, and decomposed and refined the foot model, which has been neglected by previous studies.

## 2. Materials and Methods

### 2.1. Solid Body Foot Model Preparation

The whole model (WM) is from the right foot of a male stander (Figure 1). This model is a whole foot model geometry containing the boundary surfaces of the skin [13]. The WM was required to build the skeletal structure foot model (SSM). In the right foot WM, a human right foot skeleton model was obtained and combined with the WM. Firstly, we compared the skeleton model with the overall model. Secondly, we removed the model area that overlapped with the bone model in the overall model. The final bone model overlapped with the overall model with the overlap area removed, and the release interface was defined as a whole. SolidWorks 2014 (SolidWorks Corporation, Concord, MA, USA) was used in the SSM to build each bone and the whole foot boundary [15]. In addition, human muscles and other soft tissues were defined as a whole in the SSM. The whole process is shown in Figure 2. The software was used to export the data into STEP file.

SolidWorks 2014 was used in the SSM to build each bone and the whole foot boundary [15]. In addition, human muscles and other soft tissues were defined as a whole in the SSM. The whole process is shown in Figure 2. The software was used to export the data into STEP file.

### 2.2. Properties of FEA Model

The WM’s FEA model consisted of the whole foot model and the insole model (Figure 3a), whilst the SSM’s FEA model consisted of the insole, other soft tissues of the foot without bones, and 22foot bones (e.g., talus, calcaneus, cuboid, navicular, cuneiforms and metatarsals) (Figure 3b). The WM and SSM were meshed in ANSYS 19.2(ANSYS Corporation, Pittsburgh, PA, USA) after virtual topology using the packages with a range of solid elements in the software. Virtual topology could efficiently and reasonably redivide the surface of the model without changing the geometry, thereby considerably improving the efficiency and accuracy of the simulation.

The mesh unit size was adjusted to a hexagonal mesh of 0.02 mm in size. The details of the mesh are shown in Table 1.

The automated surface-to-surface contact option in ANSYS was used to simulate the frictionless contact between the bone surfaces. When multiple models were involved in the simulation (which included the soft tissue model and insole model in the case of WM and soft tissue model, bone model, and insole model in the case of SSM), all models had to be separately connected into one system. For the contact conditions to be adopted, the bones were bonded, the bones and feet were separated by the no-separation type, and the tangential direction allowed a small slip. This type was relevant to the actual situation of the human body. The contact between insoles with foot plantar was defined by a body type.

### 2.3. Material Properties of the FEA Model

The prepared model was imported into the example created in ANSYS 19.2 software. The properties of the original materials, elastomer (Ogden shown as Figure 4), and EVA were used for the foot soft tissue and insole. The details of the material parameters used in the model are shown in Table 2 [17]. The bone material was prepared on the basis of 3D finite element studies of the relevant foot bone, and the elastic modulus was prepared. This modulus was 10 GPa, and its Poisson’s ratio was 0.34.

### 2.4. Loading Condition of the FEA Model

The human body data determined at the time of model establishment were selected to compare the effects of the two models on plantar pressure, and the weight of 60 kg was chosen as the overall load [18]. This weight was chosen to act on the cross section of the tibia and fibula, and the downward force was representative of the body’s own gravity. Given that humans have two feet under their own weight, the one foot was 30 kg. Thus, the force acting on a single foot was set to 300 N [15]. Figure 5 shows the simulation of the two models in detail. Figure 5a,b shows that the force acted on the whole cross section of the tibia and fibula in the WM, whilst the force only acted on the bone’s cross section of the tibia and fibula in the SSM.

## 3. Results

### 3.1. Analysis of Deformation and Stresses of StepEase™ Footwear

This study established two 3D finite element human foot models with precise geometrical features. Both models were capable of predicting the distribution of plantar pressure formed by the body’s own load, the stress of the foot surface, and the stress/strain of the footwear. By comparing the experimental results of the two models, researchers could obtain in-depth understandings of the effects of bone on the stress and strain of the foot and footwear by using the foot model.

Figure 6 depicts the plantar pressure distribution obtained from the WM under 300 N loading. The model’s total deformation, insole deformation, plantar pressures, and insole stresses are shown in Figure 6a–d, respectively. Figure 6 shows that the maximum total and insole deformations were 1.46 × 10^−2^ and 1.14 × 10^−5^ m, respectively, in the WM under 300 N force on the ankle cross section. Under the same condition, the maximum stresses of foot plantar and insole were 1.94 × 10^5^ and 3.86 × 10^5^ Pa, respectively.

Figure 7 depicts the plantar pressure distribution obtained from the SSM under 300 N loading. The model’s total deformation, insole deformation, plantar pressures, and insole stresses are demonstrated in Figure 7a–d, respectively. Figure 7 indicates that the maximum total and insole deformations were 1.48 and 7.97 × 10^−4^ cm in the SSM under 300 N force on the ankle cross section. Under the same condition, the maximum stresses of foot plantar and insole were 1.86 × 10^5^ and 2.99 × 10^5^ Pa, respectively.

The comparison of the simulation results of the WM and the SSM clearly showed whether the case was total deformation, insole deformation, plantar pressures, or insole stresses. The SSM was higher than the WM at the maximum. The SSM was 1.35%, 43.04%, 4.30%, and 29.10% higher than the WM in terms of total deformation, insole deformation, plantar pressures, and insole stresses, respectively.

The stress accumulation in these places could cause ulcers via the rupture of human skin tissue. This finding was consistent with the actual location of foot ulcers in patients with diabetes [17,19,20]. Figure 6 and Figure 7 and Table 3 show that the stresses of the foot focused on the forefoot and heel, and no excessive stress distribution was found in the part of the foot, which was consistent with the human foot structure. The WM and SSM’s plantar pressure are shown in Figure 8, and compared with that obtained from literature to validate the simulation, as shown in Table 4.

The peak of the diabetic foot and foot pressure obtained using FEA was basically consistent with that reported in the literature [13,15,16]. For example, Taha et al. [15] performed a finite element analysis of the human foot model to study the dynamic behavior and internal load conditions on a plane ground, and the peak of the plantar stress was 1.60 × 10^−1^ MPa. Cheung et al. [16] studied the effect of the material hardness of the insole on the plantar pressure and stress distribution of the bone and ligament structures during balanced standing. Their results indicated that the peak of the plantar pressure was 1.70 × 10^−1^ MPa. Qui et al. [21] reported that the peak of the plantar pressure, which was closest to that in the present study, was 1.98 × 10^−1^ MPa, and the plantar pressure was mainly distributed at the heel. However, the stress distribution in other parts was different. This foot pressure peak had a certain gap with that from the present study, which may be due to the difference in bone model and contact conditions.

The overall deformation included the deformation of the insole and that of the foot model. The deformation of the insole proportionally occupied a major part of the total deformation, which could be found in the material properties of the insole. This deformation was much smaller than that in the rest of the model in terms of Young’s modulus, which could be found in the details of the material properties in Table 2 [13]. It considerably reduces the deformation of the human foot and plantar pressure by increasing the deformation of the insole [17]. This phenomenon could be observed in the deformation of the deformed cloud image in the total deformation from the Internet. The main factor affecting stress distribution is unclear compared with that affecting deformation. This finding showed that the simulation results matched the actual results in Table 3. The stress cloud diagram in Table 3 indicates that the distribution of plantar pressure was relatively discrete due to the material properties of the insoles. This result may be due to the fact that when the insole material (EVA foam) receives pressure, this pressure is substantially converted into the displacement and deformation of the particles, thereby reducing stress.

However, whether it is observed from stress or deformation in Figure 5 and Figure 6, the results of the two models were compared. The plantar pressure and insole deformation in the WM were larger than those in the SSM. A slight difference in total deformation was found between the WM (1.46 cm) and the SSM (1.48 cm). The deformation of the SSM was 1.35% higher than that of the WM; hence, it could be ignored.

However, a huge difference in insole deformation was observed between the two models, possibly because of the presence of a skeletal structure, the material of which was harder than soft tissue and could transmit force more easily. Therefore, in the SSM, when force acted on the bone structure, this structure did not produce a large deformation but a more complete force transmission. Thus, in the SSM, the force in the plantar tissue was bigger than that in the WM in the same area. The SSM produced a deformation at the bottom of the foot that was much larger than that in the WM in the same position. The total deformation in the two models was similar, and the insole deformation in the WM was larger than that in the SSM. The SSM produced a deformation at the bottom of the foot that was much larger than that at the bottom of the WM foot. The final FE results showed that the insole deformation of the WM was bigger than that of the SSM by 43.04%.

The deformation in the FE results was analyzed, and the finding showed that the insole deformation in the SSM was smaller than that in the WM. The stress–strain curve of the EVA insole material demonstrated that the corresponding point of the insole in the WM was larger than that in the SSM, indicating that the hardness of the insole in the WM was greater than that in the SSM, which also showed the hardness of the plantar tissue in previous studies [13]. The increase in hardness could cause an increase in the plantar stress of the interaction [17].

The increase in hardness of the contact interface resulted in increased plantar stress, which explained why the WM had greater foot pressure than the SSM and why the WM insole pressure was much larger than the SSM insole pressure. Previous articles have shown that the plantar pressure could be reduced by increasing the deformation [15,21]. Thus, the deformation in the insole of the SSM was bigger than that of the WM, and the plantar pressure of the SSM was smaller than that of the WM.

### 3.2. Mathematical Model

On the basis of the study of the deformation and stress in StepEase™ [1] shoe, further research was conducted to determine the sole thickness using EVA to reduce the plantar pressure distribution. The patient’s body weight was divided into 10 kg, and the relationship between body weight in the range of 60–110 kg and sole thickness in the range of 0.5–1.5 cm was compared. The experimental results of the two models were compared with those [15] of the pressure distribution when the Tekscan and CT scan machines were applied to the patient’s upright local time.

Table 4 and Table 5 show the results of previous studies of pressure distributions in the two models of the foot contour of patients with diabetes, which were classified and simulated on the basis of the thickness of the EVA insole

Table 3 shows the relationship between patient weight and peak pressure in the WM. The peak pressure was the lowest at 1 cm under 60 and 70 kg body weights, and the peak pressure at 1.5 cm was under 80, 90, 100, and 110 kg body weights. Table 5 shows the relationship between patient weight and peak pressure in the SSM. In the WM, the peak pressure was the lowest at 0.75 cm under 60 kg body weight, and the peak pressure at 1.5 cm was under 70, 80, 90, 100, and 110 kg body weights. The insole thickness was chosen by comparing the maximum pressure distribution of each patient’s body weight with its corresponding peak pressure. The decrease in pressure distribution was not proportional to the increase in insole thickness. High weight does not always require a large amount of thickness to reduce the pressure distribution. The observation results demonstrated that the plantar pressure distribution could be effectively reduced whether or not thickness was changed in accordance with the patient’s body weight in the WM or the SSM. Through this discovery, developing a corresponding mathematical model could promote and determine the optimal insole thickness for optimal reduction of plantar pressure.

In accordance with the data in Table 4 and Table 5, the corresponding figures were generated in Figure 9a to obtain a mathematical model. The line in the figure indicated that the patient’s weight was proportional to the appropriate insole thickness. The three curves in the figure represented the experimental results of the predecessors [16], the FEA results of the WM, and the FEA results of the SSM. The corresponding mathematical models were as follows: experiment, y = 0.005714x + 0.681; SSM, y = 0.01071x + 0.4643; and WM, y = 0.01143x + 0.3619. Through this model, whether the results of the finite element simulation were similar to those obtained by actual experiments could be determined. In addition, the mathematical model could obtain the appropriate insole thickness corresponding to the patient’s body weight. After the mathematical model was obtained, different patient weights were substituted into the model to calculate the corresponding suitable insole thickness and compared with the data obtained in the experiment [1]. The final result is shown in Table 6.

The data indicated that the thickness estimates based on the mathematical model showed the difference in patient weight in each category. The results of the mathematical model of the SSM and the WM for lightweight patients were close to the experimental results. As the weight increased, a certain error in the results was obtained by either the SSM or the mathematical model of the WM, possibly because the gradient factor on the graph randomly affected the value of the mathematical model. In addition, with the increase in body weight, the physical properties of the EVA foam were abruptly changed in the actual experiment. However, this change did not occur in the simulation.

The simulation results were divided and solved using the linear method. As the patient’s weight increased, its peak pressure approximated a proportional increase, causing the simulation result to have larger error with the increase in the variable. Figure 9a illustrates the line of the WM and the SSM. With the increase in patient’s weight, the differences between them and the experiment were increased, which may require a correction factor to correct the mathematical model to some extent. The other fit could be considered to investigate the relationship between patient weight and adaptive insole thickness, such as that in Figure 9b and Table 7.

The functions corresponding to the three curves obtained by fitting in the figure are shown. The thickness corresponding to the different body weights in the fitting curve was very close to the sample point (the specific values are shown in Table 5), indicating that the curve had a high fitting precision. This curve was compared with that in Figure 9a. The results showed that this curve could obviously reflect the relationship between body weight and insole thickness.

## 4. Conclusions

The EVA polymer could be used to decrease the risk of diabetic foot ulcers in patients with diabetes. The weight, size, and foot form of patients are key factors for developing a suitable insole to be used for patients. In this study, finite element methods were used to determine the effects of two different models on footwear and build a mathematical model to determine the insole thickness using EVA to reduce the plantar pressure distribution. The results of the 3D WM and SSM and the corresponding finite element model for simulation showed the model’s total deformation, insole deformation, plantar pressures, and insole stresses. The results were consistent with actual human body structure. The simulation results showed that the EVA insole could have a certain effect on plantar pressure. Adding the skeletal structure caused a 1.35%, 43.04%, 4.30%, and 29.10% reduction in total deformation, insole deformation, plantar pressures, and insole stresses, respectively. The effect of EVA insole on sole pressure was explored. Adding the skeletal structure further increased the accuracy of the simulation results and considerably improved the authenticity of the simulation. The mathematical model could provide strong technical support and guarantee for the industrial production of EVA insole. Thus, we can produce footwear with cost-effective manufacturing using this technology.

## Figures and Tables

**Figure 1 polymers-13-02261-f001:**
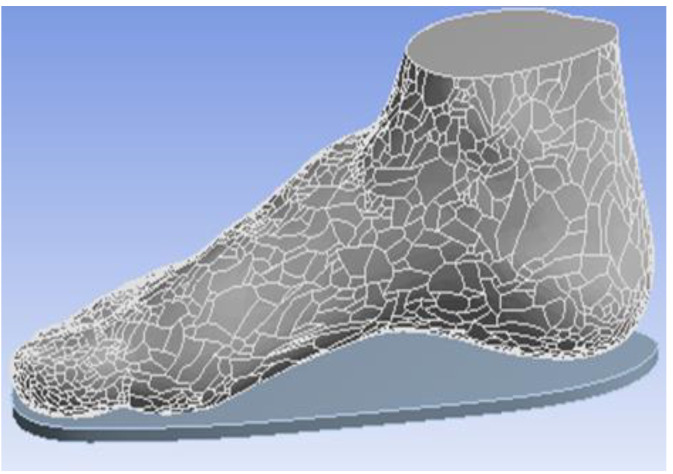
Whole foot model geometry.

**Figure 2 polymers-13-02261-f002:**
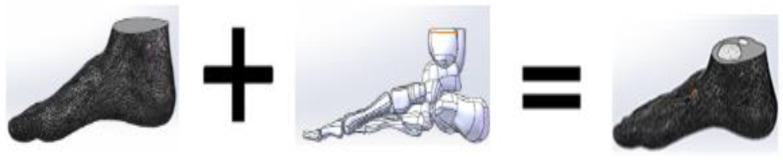
Model construction diagram.

**Figure 3 polymers-13-02261-f003:**
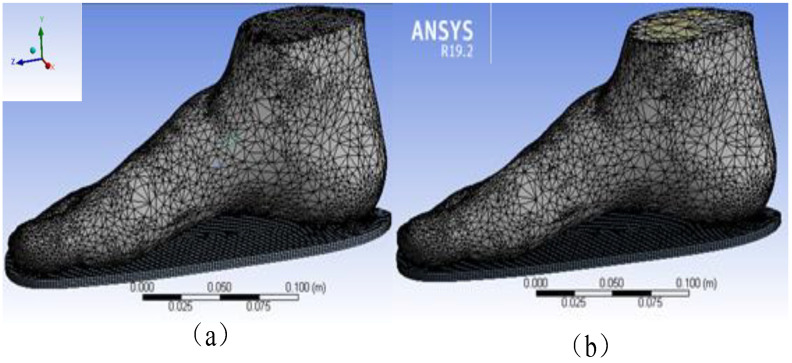
Finite element analysis (FEA) model of (**a**) the whole model (WM) and (**b**) the skeletal structure foot model (SSM).

**Figure 4 polymers-13-02261-f004:**
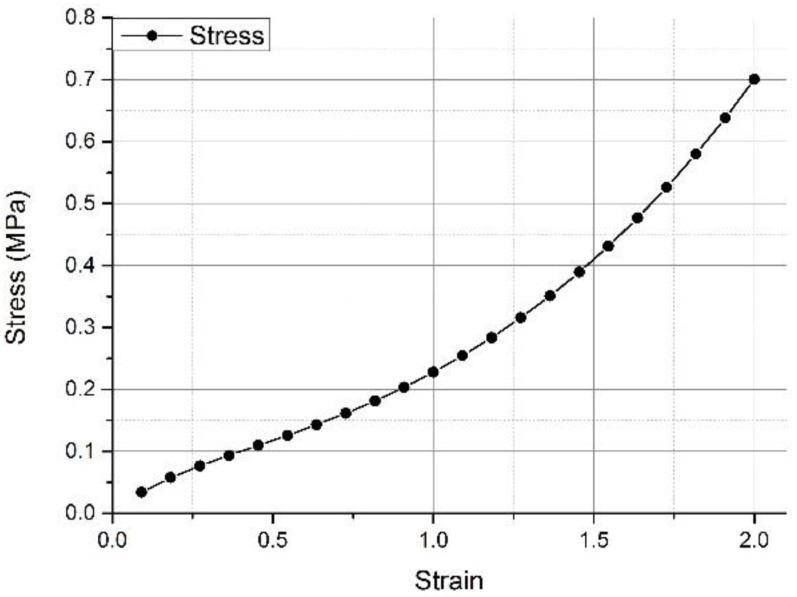
Nonlinear stress–strain response of soft tissue adopted for the FEA model [13].

**Figure 5 polymers-13-02261-f005:**
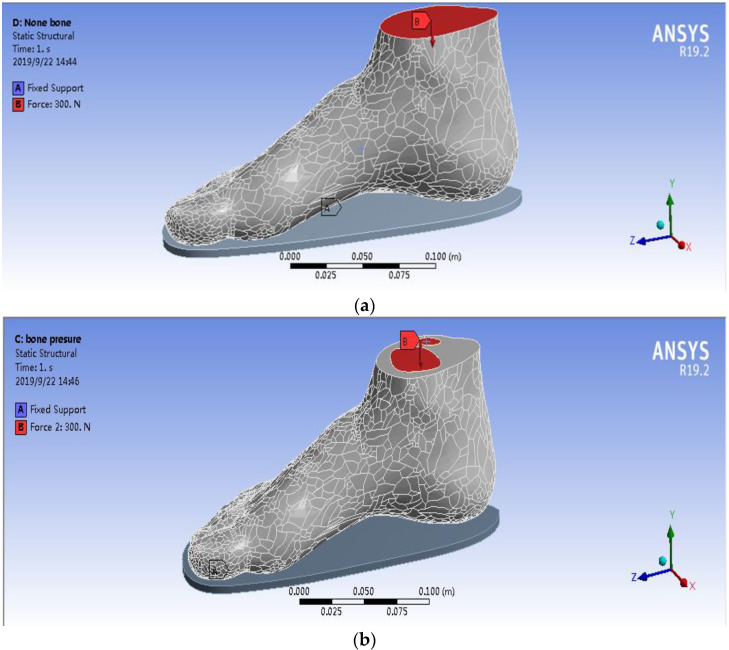
Loading definition of the FEA model. (**a**) WM and (**b**) SSM.

**Figure 6 polymers-13-02261-f006:**
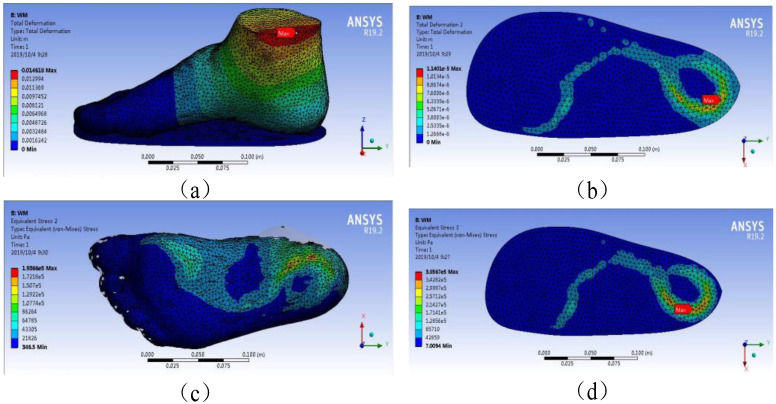
Plantar pressure distribution obtained from WM under 300 N loading. (**a**) model’s total deformation, (**b**) insole deformation, (**c**) plantar pressures, (**d**) insole stresses.

**Figure 7 polymers-13-02261-f007:**
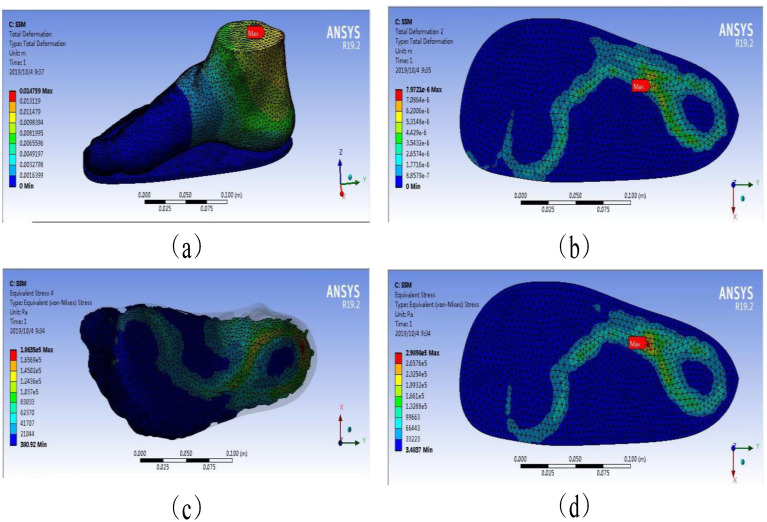
Plantar pressure distribution obtained from SSM under 300 N loading. (**a**) model’s total deformation, (**b**) insole deformation, (**c**) plantar pressures, (**d**) insole stresses.

**Figure 8 polymers-13-02261-f008:**
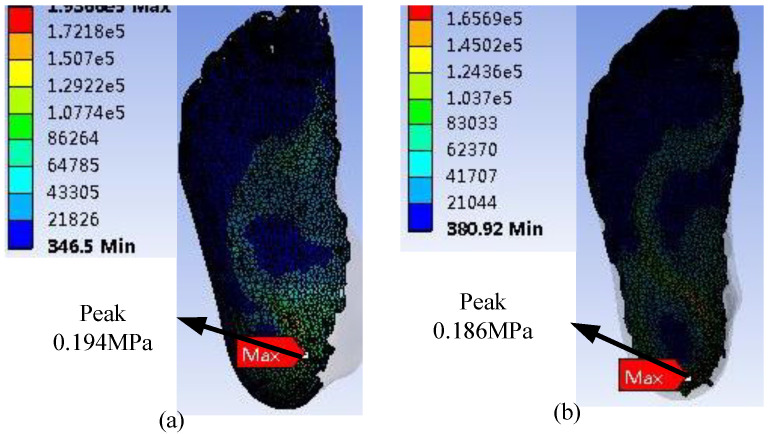
Plantar pressure. (**a**) WM and (**b**) SSM.

**Figure 9 polymers-13-02261-f009:**
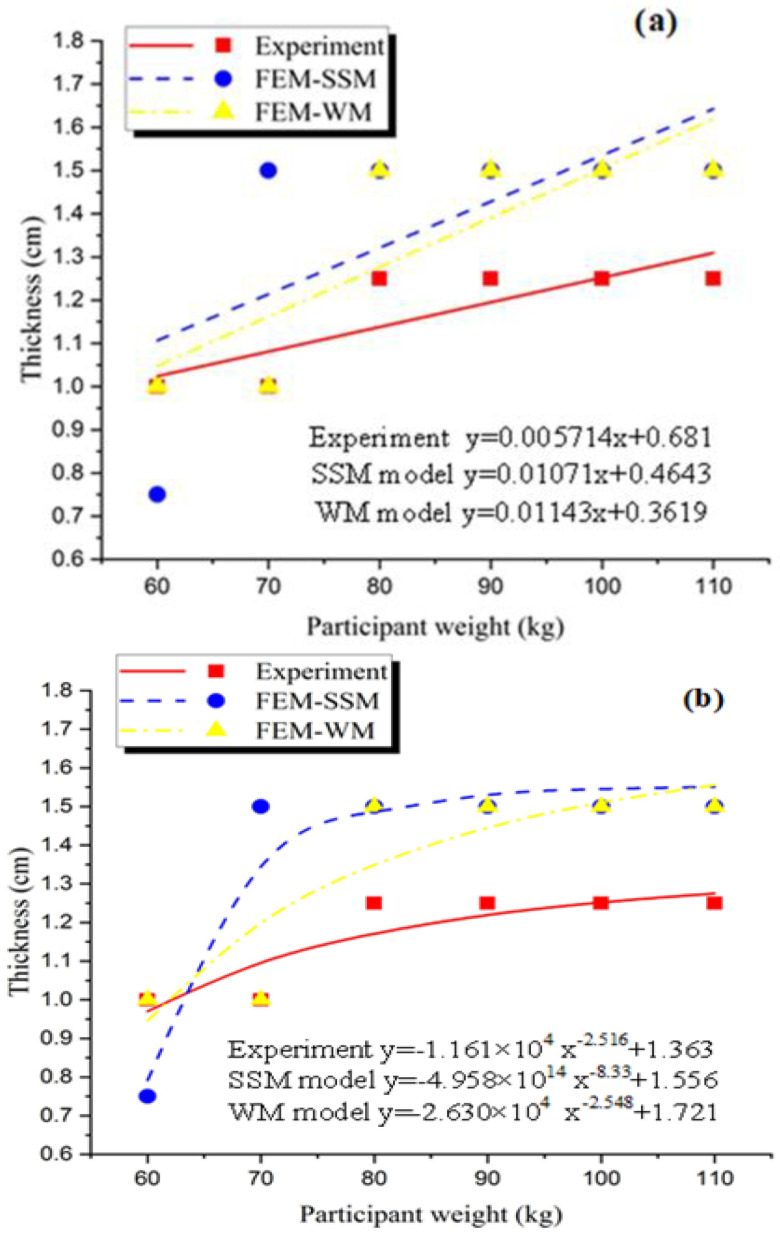
Graph between the weight of the patient against the appropriate thickness; (**a**) is polynomial and (**b**) is power.

**Table 1 polymers-13-02261-t001:** Element types of the finite element model.

	Element	Node	Element Type		Virtual Faces	Virtual Edge
			Insole	Others		
WM model	59,655	163,599	Hexahedron mesh	Tetrahedral mesh	2675	36
SSM model	72,214	187,473	Hexahedron mesh	Tetrahedral mesh	2832	201

**Table 2 polymers-13-02261-t002:** Details of the material properties [13].

	Elastic Modulus (MPa)	Poisson Ratio
Foot soft tissue	1.15 × 10^−6^	0.49
Bone	10,000	0.34
Footwear	200	0.4

**Table 3 polymers-13-02261-t003:** Data on the relationship between different insole thicknesses and plantar pressures under human body weight in the WM.

Patient Weight (kg)	Optimal Thickness of Experiment (cm)	Optimal Thickness of Experiment Mathematical Model (cm)	Optimal Thickness of SSM (cm)	Optimal Thickness of SSM Mathematical Model (cm)	Optimal Thickness of WM (cm)	Optimal Thickness of WM Mathematical Model (cm)
60	1	1.0238	0.75	1.1069	1	1.0477
70	1	1.081	1.5	1.214	1	1.162
80	1.25	1.1381	1.5	1.3211	1.5	1.2763
90	1.25	1.1953	1.5	1.4282	1.5	1.3906
100	1.25	1.2524	1.5	1.5353	1.5	1.5049
110	1.25	1.3095	1.5	1.6424	1.5	1.6192

**Table 4 polymers-13-02261-t004:** Validation of plantar pressure from FEA.

Plantar Pressure from FEA		Plantar Pressure from Literatures		
WM model	SSM model	Qui et al. 2011 [21]	Taha et al. 2016 [15]	Cheung et al. 2005 [16]
Peak 1.94 × 10^−1^ MPa	Peak 1.86 × 10^−1^ MPa	Peak 1.98 × 10^−1^ MPa	Peak 1.60 × 10^−1^ MPa	Peak 1.70 × 10^−1^ MPa

**Table 5 polymers-13-02261-t005:** Data on the relationship between different insole thicknesses and plantar pressures under human body weight in the SSM.

Patient Weight (kg)	Optimal Thickness of Experiment (cm)	Optimal Thickness of Experiment Mathematical Model (cm)	Optimal Thickness of SSM (cm)	Optimal Thickness of SSM Mathematical Model (cm)	Optimal Thickness of WM (cm)	Optimal Thickness of WM Mathematical Model (cm)
60	1	0.9701	0.75	0.7916	1	0.9461
70	1	1.0954	1.5	1.3443	1	1.1978
80	1.25	1.1709	1.5	1.4864	1.5	1.3487
90	1.25	1.2194	1.5	1.5299	1.5	1.4452
100	1.25	1.2521	1.5	1.5452	1.5	1.5102
110	1.25	1.2751	1.5	1.5511	1.5	1.5556

**Table 6 polymers-13-02261-t006:** Comparison between thicknesses based on mathematical model by polynomial fit.

Patient Weight (kg)	Optimal Thickness of Experiment (cm)	Optimal Thickness of Experiment Mathematical Model (cm)	Optimal Thickness of SSM (cm)	Optimal Thickness of SSM Mathematical Model (cm)	Optimal Thickness of WM (cm)	Optimal Thickness of WM Mathematical Model (cm)
60	1	1.0238	0.75	1.1069	1	1.0477
70	1	1.081	1.5	1.214	1	1.162
80	1.25	1.1381	1.5	1.3211	1.5	1.2763
90	1.25	1.1953	1.5	1.4282	1.5	1.3906
100	1.25	1.2524	1.5	1.5353	1.5	1.5049
110	1.25	1.3095	1.5	1.6424	1.5	1.6192

**Table 7 polymers-13-02261-t007:** Comparison between thicknesses based on mathematical model by power fit.

Patient Weight (kg)	Optimal Thickness of Experiment (cm)	Optimal Thickness of Experiment Mathematical Model (cm)	Optimal Thickness of SSM (cm)	Optimal Thickness of SSM Mathematical Model (cm)	Optimal Thickness of WM (cm)	Optimal Thickness of WM Mathematical Model (cm)
60	1	0.9701	0.75	0.7916	1	0.9461
70	1	1.0954	1.5	1.3443	1	1.1978
80	1.25	1.1709	1.5	1.4864	1.5	1.3487
90	1.25	1.2194	1.5	1.5299	1.5	1.4452
100	1.25	1.2521	1.5	1.5452	1.5	1.5102
110	1.25	1.2751	1.5	1.5511	1.5	1.5556

## Data Availability

The data presented in this study are available on request from the corresponding author.

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
