# Peer review of "Finite Element Analysis of Cushioned Diabetic Footwear Using Ethylene Vinyl Acetate Polymer"

_polymers, 2021, doi:10.3390/polym13142261_

Round 1
Reviewer 1 Report
This manuscript entitled “Finite element analysis of cushioned diabetic footwear using ethylene vinyl acetate polymer” aimed to evaluate the stress-strain of EVA cushioning insole on diabetic patients using the FEA approach. Although it proposed a technique framework combined with wearable sensors for running protocol evaluation, drawbacks may limit this manuscript to the standard of being published. The abstract presented too much background information but did not show decent methods and findings. In the introduction section, poor academic writing, and lack of logic in the literature review should be notified and make revisions. The methodologies missed details for instance how to build the skeletal structure foot model? Why this study compared the WM and SSM models, as the more precise and authentic model is no doubt can improve the simulation accuracy. How to obtain soft tissue model without bones? Were the mesh sizes of bone and soft tissue being the same? The results and discussion are mixed, and the main findings didn’t present clearly.
Some suggestions are listed in the specific comments below.
Specific comments:
- Line2 15-16, change ‘redistribute plantar pressure and relieve plantar pressure’ to ‘redistribute and relieve plantar pressure’.
- Line 18-19, considering remove ‘in the human foot model, human bones are an objective existence that could not be ignored’.
- Lines 21-22, ‘to explore the influence of human bones on the plantar pressure and obtain accurate foot pressure.’
- Please notes the citation in lines 45 and 57.
- The manuscript should be amended by a native English professor.
- Line 86, how the combine skeleton model with WM?
- Lines 89-92, how to create and build each foot bones? Was it drafted from a sketch or scanned from a medical image (i.e. CT).
- What was the boundary condition in SSM and how to fix it?
- Please specific (a) and (b) in figure 5.
- replace the unit to cm, i.e. 1.48 cm and 7.97×10-4 cm in line 165.
- In table 3, plantar pressure from literature also measured pressure using FEA, so subtitles cannot distinguish two categories.
- In the conclusion, it is no doubt ‘3. Adding the skeleton structure further increased the accuracy of the simulation results and considerably improved the authenticity of the simulation.’
Author Response
Dear Editor,
We are grateful for the opportunity to resubmit our manuscript entitled, “Finite Element Analysis of Cushioned Diabetic Footwear Using Ethylene Vinyl Acetate Polymer “. We thank the reviewers for providing insightful comments to improve our manuscript. As below, on behalf of my co-author, I would like to clarify some of the points raised by the Reviewer. And we hope the Reviewer and the Editor will be satisfied with our responses to the ‘comments’ and the revisions for the original manuscript. The imposed changes are in blue colour in original paper.
In response to the questions raised by the experts, our team conducted detailed discussions and analyses. We have further organized and modified our current work. First of all, the establishment of the skeleton model in this article is obtained by referring to previous studies, and this skeleton model and soft tissue model are obtained by CT tomography of the actual human foot. Secondly, the main content of the research in this article, as experts said, is to compare the difference between the whole model (WM) and the bone soft tissue model. Therefore, what needs to be guaranteed for this article is the consistency between the two sets of models, so the grid types and grid sizes used by the two sets of models are the same. It also ensures that the mesh size of the bones and soft tissues is consistent.
|
Question 1& Respond1 |
Line2 15-16, change ‘redistribute plantar pressure and relieve plantar pressure’ to ‘redistribute and relieve plantar pressure’.
Respond: Already done
|
Question 2 & Respond2 |
. Line 18-19, considering remove ‘in the human foot model, human bones are an objective existence that could not be ignored’
Respond: Already done
|
Question 3 & Respond 3 |
Lines 21-22, ‘to explore the influence of human bones on the plantar pressure and obtain accurate foot pressure.
Respond: Already done
|
Question 4& Respond 4 |
4- Please notes the citation in lines 45 and 57
Respond: Line 45 was amended and Ref 15, was added in line 57; Taha, Z.; Norman, M.S.; Omar, S.F.S.; Suwarganda, E. A finite element analysis of a human foot model to simulate neutral standing on ground. Procedia engineering 2016, 147, 240-245.
|
Question 5& Respond 5 |
The manuscript should be amended by a native English professor.
Respond: In response to the questions raised by the experts, the full text was read and revised in detail by members of the team whose native language is English.
|
Question 6& Respond 6 |
Line 86, how the combine skeleton model with WM?
Respond: First, compare the skeleton model with the overall model. Secondly, remove the model area that overlaps with the bone model in the overall model. The final bone model overlaps with the overall model with the overlap area removed, and the release interface is defined as a whole.
|
Question 7& Respond 7 |
Lines 89-92, how to create and build each foot bones? Was it drafted from a sketch or scanned from a medical image (i.e. CT).
Respond: The footstep bone model is obtained by CT scan.
|
Question 8& Respond 8 |
What was the boundary condition in SSM and how to fix it?
Respond: The determination of the boundary conditions of SSM is mainly based on the WM model. In order to compare the validity and scientificity of the results, the boundary conditions at the bottom of the foot are the same in the SSM and WM models. The bone and soft tissue are determined as an organic whole, so the boundary conditions of the bone and soft tissue are determined as a fixed pattern. At the same time, the SSM and WM models use the same fixing method.
|
Question 9& Respond 9 |
Please specific (a) and (b) in figure 5.
Response: Loading definition of the FEA model. (a) WM and (b) SSM
|
Question 10& Respond 10 |
Replace the unit to cm, i.e. 1.48 cm and 7.97×10-4 cm in line 165.
Response: Already dome
|
Question 11& Respond 11 |
In table 3, plantar pressure from literature also measured pressure using FEA, so subtitles cannot distinguish two categories.
Response: The proofreading was already done
|
Question 12& Respond 12 |
In the conclusion, it is no doubt ‘3. Adding the skeleton structure further increased the accuracy of the simulation results and considerably improved the authenticity of the simulation.
Response: The proofreading was already done

Reviewer 2 Report
This paper applied finite element methods to determine the effects of two different models on footwear and developed a mathematical model to evaluate the insole thickness using EVA to reduce the plantar pressure distribution. I have the following comments that I would like to be addressed by the authors.
1. Please highlight the novelty of the proposed method in the abstract and introduction. Also, the literature review in the introduction should be elaborated by including more recent work.
2. There are many linguistic and grammatical typos, such as page 1 line 44, “Cheung et al (2005), studied …”, missing the reference number. Page 2, line 57, “[15] performed a finite element analysis...” is not a complete sentence. Page 2, line 85, “skeletal structure foot model (SSM)” should be SSFM instead of SSM. There are many other grammatical typos like this. Please carefully read through and conduct the proofreading.
3. Page 2, line 78, what is ‘StepEase’, please add detailed information.
5. The table caption format of Table 4 is not correct; please modify it based on the format of Polymers.
6. Regarding the finite element model (FEM), the paper does not report any aspect associated with the element type and convergence of the FEM model. The convergence is a critical issue in terms of the quality of the solution, and in terms of the computational load associated with the analysis. Please include the convergence study of the FEM and the convergence criteria in your analysis.
7. Figure 8, comparison between FEM result and experiment. There is a large gap between prediction and experiment, which means that the FEM analysis is not accurate based on the presented results. Please comment and elaborate on it.
8. At the end of the manuscript, please describe the scheme of the intended application of the developed method in real practice. What conditions must be met? What preliminary analysis should be carried out? What is the expected performance of this method? What are the limitations of this method on the examined structures and conditions of work?
Author Response
Dear Editor,
We are grateful for the opportunity to resubmit our manuscript entitled, “Finite Element Analysis of Cushioned Diabetic Footwear Using Ethylene Vinyl Acetate Polymer “. We thank the reviewers for providing insightful comments to improve our manuscript. As below, on behalf of my co-author, I would like to clarify some of the points raised by the Reviewer. And we hope the Reviewer and the Editor will be satisfied with our responses to the ‘comments’ and the revisions for the original manuscript. The imposed changes are in blue colour in original paper
|
Question 1& Respond1 |
Please highlight the novelty of the proposed method in the abstract and introduction. Also, the literature review in the introduction should be elaborated by including more recent work.
Respond: In response to the opinions of experts, the abstract and introduction focus on the new research ideas put forward in this article, and the novelty of this article is further described in more detail. Compared with traditional research work, this article first puts forward the idea of separating the bones and soft tissues of the foot, and decomposes and refines the foot model. This is not considered by the predecessors
|
Question 2& Respond2 |
There are many linguistic and grammatical typos, such as page 1 line 44, “Cheung et al (2005), studied …”, missing the reference number. Page 2, line 57, “[15] performed a finite element analysis...” is not a complete sentence. Page 2, line 85, “skeletal structure foot model (SSM)” should be SSFM instead of SSM. There are many other grammatical typos like this. Please carefully read through and conduct the proofreading.
Respond: In response to the language errors of this article, the team members repeatedly reviewed the manuscript through multiple people. It was finally reviewed and revised by English-speaking members
|
Question 3& Respond3 |
Page 2, line 78, what is ‘StepEase’, please add detailed information.
Respond: Thank you very much for the questions raised by the experts, due to the repeated revisions of the manuscript. The original ‘StepEaseTM’ in the article has become StepEase, and all related content has been modified. StepEase is a registered trademark, neither a name nor an advertisement
|
Question 4& Respond4 |
The table caption format of Table 4 is not correct; please modify it based on the format of Polymers.
Respond: In response to questions raised by experts, revisions were made. And the name format of all tables and pictures in the text has been revised.
|
Question 5& Respond5 |
Regarding the finite element model (FEM), the paper does not report any aspect associated with the element type and convergence of the FEM model. The convergence is a critical issue in terms of the quality of the solution, and in terms of the computational load associated with the analysis. Please include the convergence study of the FEM and the convergence criteria in your analysis.
Respond: The establishment of the FEM model and the related information of the grid element types are described in the chapters and tables of the model establishment. The mathematical solution of the ANSYS software used in this paper is related to the configuration, so the convergence of the solution is guaranteed. The focus of this article is on different models, including but not limited to (model boundary conditions, element types, etc.) under the same conditions, so this article explains less about this aspect. At the same time, in order to ensure that the result is convergent, it has a certain degree of scientificity and accuracy, so the final simulation result is horizontally compared with the relevant literature. To further confirm the feasibility and authenticity of the overall simulation process, so the convergence of the model is not described in detail unilaterally.
|
Question 6& Respond6 |
Figure 8, comparison between FEM result and experiment. There is a large gap between prediction and experiment, which means that the FEM analysis is not accurate based on the presented results. Please comment and elaborate on it.
Respond: The simulation results and the experimental results do have certain errors, but they are not as big as the gap between the points in Figure 8. The three different points shown in Figure 8 are not derived from direct FEM results. The simulation results obtained by FEM are the contact pressures corresponding to footwear sizes of 0.5cm, 0.75cm, 1cm and equal thickness under different weights, by selecting the size and thickness corresponding to the smallest contact pressure. Therefore, this magnifies the error. In other words, even if the contact pressure obtained by the simulation result is very different from the actual contact pressure, the different thickness and size intervals cause a big difference.
|
Question 7& Respond7 |
At the end of the manuscript, please describe the scheme of the intended application of the developed method in real practice. What conditions must be met? What preliminary analysis should be carried out? What is the expected performance of this method? What are the limitations of this method on the examined structures and conditions of work?
Respond: Through your comment, you can see that you have a full understanding of our work, and you are an expert in this area. The actual application plan proposed at the end of this article is only a preliminary direction, and its actual operation and application still have a lot of work to be determined as described by experts. As far as the practical application mentioned in this article is concerned, it does have certain limitations. The condition that it must meet is that its weight is within the range of the sample studied in this article, and its foot size and physiological state are also within the healthy range. . For the actual use of the method described at the end of this article, a simple assessment of the user's feet is required to measure their weight. This method is expected to quickly provide the best footwear styles and types corresponding to different users. This is also our future research direction, and it is where we need to work hard in the future.

Round 2
Reviewer 1 Report
Authors have now addressed the issues I had for the previous manuscript. However, the language should be polished to improve the readability of this manuscript and assist understanding of this study.
Author Response
Authors have now addressed the issues I had for the previous manuscript. However, the language should be polished to improve the readability of this manuscript and assist understanding of this study.
Response: Thank you very much for reviewing and evaluating this manuscript during your busy schedule, and for reading our revised comments in detail. The manuscript was amended by two native English speakers as attached file.

Reviewer 2 Report
After the revision, most of my comments have been addressed. However, the authors should more discussions at the end of the manuscript. Please carefully read through the manuscript and conduct the proofreading.
Author Response
After the revision, most of my comments have been addressed. However, the authors should more discussions at the end of the manuscript. Please carefully read through the manuscript and conduct the proofreading.
Response: Thank you very much for reviewing and evaluating this manuscript during your busy schedule, and for reading our revised comments in detail. We tried to improve the conclusion. So, the imposed changes are in blue colour in original paper. Furthermore, the manuscript was amended by two native English speakers attached file.